# Using the Theory of Planned Behavior and Past Behavior to Explain the Intention to Receive a Seasonal Influenza Vaccine among Family Caregivers of People with Dementia

Francesco Bruno [1,*], Paolo Abondio [2,3,*], Valentina Laganà [1], Rosanna Colao [4], Sabrina M. Curcio [4], Francesca Frangipane [4], Gianfranco Puccio [4], Raffaele Di Lorenzo [1], Amalia C. Bruni [4,†] and Raffaele Maletta [4]

1    Association for Neurogenetic Research (ARN), 88046 Lamezia Terme, Italy
2    Laboratory of Ancient DNA, Department of Cultural Heritage, University of Bologna, 48121 Ravenna, Italy
3    Laboratory of Molecular Anthropology and Center for Genome Biology, Department of Biological, Geological and Environmental Sciences, University of Bologna, 40126 Bologna, Italy
4    Regional Neurogenetic Centre (CRN), Department of Primary Care, ASP Catanzaro, 88046 Lamezia Terme, Italy
*    Correspondence: francescobrunofb@gmail.com (F.B.); paolo.abondio2@unibo.it (P.A.)
†    A.C.B. is former Director of the Regional Neurogenetic Centre (CRN).

**Abstract:** Older adults with dementia present an increased risk of mortality due to seasonal influenza. Despite concerning evidence, the influenza vaccination program has been unsuccessful, with low rates of uptake in Italian people ≥65 years. In addition, being vaccinated does not eliminate the risk of contracting a virus, especially by coming into close contact with other possibly unvaccinated people, such as family caregivers in the home environment. Therefore, the refusal of family caregivers to get vaccinated for seasonal influenza could have dire consequences for their relatives with dementia. The aims of this study were to investigate the predictive role of the Theory of Planned Behavior model (TPB) and past vaccination behavior on the intention to receive a seasonal influenza vaccine among family caregivers of people with dementia. Data were collected from seventy-one respondents during July–September 2021 using a cross-sectional web-based survey design. Results of hierarchical binary logistic regression showed that TPB (i.e., attitudes towards vaccination, subjective norms, and perceived behavioral control) explained 51.6% of the variance in intention to receive a seasonal influenza vaccine; past vaccination behavior increased this to 58.8%. In conclusion, past vaccination behavior and the theory of planned behavior variables effectively predict influenza vaccine willingness of family caregivers of people with dementia and should be targeted in vaccination campaigns.

**Keywords:** family caregivers; dementia; theory of planned behavior; seasonal influenza; vaccine hesitancy; vaccination intentions; attitudes towards vaccinations; perceived behavioral control; subjective norms; past vaccination behavior

## 1. Introduction

Dementias represent a complex of chronic-degenerative pathologies characterized by progressive cognitive deficits [1]—such as memory, language, attention, and visuospatial and executive function impairments—behavioral disturbances [2,3]—such as apathy, depression, irritability, agitation, aggression, delusions, and hallucinations—and functional decline [4]. These clinical features could lead to the loss of autonomy and self-sufficiency and consequent dependence on others both for basic (e.g., dressing, eating, toileting, bathing, mobility) and instrumental (e.g., managing finances, cooking and meal preparation, medical management, transportation) activities of daily life [4]. Some types of dementia are classifiable as "primary" (e.g., Alzheimer's disease, dementia with Lewy bodies, frontotemporal dementia) and others as "secondary" to other conditions (e.g., vascular dementia) [5]. Although most cases of dementia are sporadic—the causes of which have

not yet been fully understood—there are also forms with a high familial recurrence and/or with autosomal dominant transmission of causative mutations from the parent to the offspring [6–10]. The prevalence of dementia is on the rise in the general population, and this evidence prompted the World Health Organization and Alzheimer Disease International to consider dementia as a global public health priority [11].

Family caregivers are all family members who assist a relative affected by dementia. Their task is to take charge of the daily person's well-being, both from a physical (e.g., cleaning and hygiene, meals, medications), practical (e.g., providing medication, organizing medical examinations and rehabilitations), and emotional (e.g., provide support, stimulate patients to have conversations, entertain them with games, films, reading) point of view [12]. The commitment required by care can lead to the development of psychological distress (e.g., depressive symptoms) or caregiver burden [13].

Seasonal influenza is a contagious respiratory disease, caused by influenza viruses, which is contracted following contact with the saliva or respiratory secretions of infected individuals, due to coughing, sneezing or even simply the action of talking [14]. It can also be contracted via touching contaminated objects followed by hand contact with the mouth, nose, or eyes [15]. The typical symptoms of seasonal influenza include fever, cough, runny nose, sore throat, chills, and joint and muscle pain [14,16]. In some cases, especially when the virus affects infants, vomiting and diarrhea may occur. In others, especially older people, there may be weakness, fatigue, and confusion [17], with an increased hospitalization rate [18]. Rosano and colleagues [19] reported that Italy showed a higher excess mortality attributable to seasonal influenza when compared to other European countries. In addition, other studies showed that older adults with dementia tend to die more often from seasonal influenza [20]. Evidence suggests that the seasonal influenza vaccines have a moderate preventive effect among elderly people and that it significantly decreases the morbidity of seasonal influenza and pneumonia—which represent a primary cause of complications and mortality in people affected by dementia [21]—as well as the risk of hospitalization and death (World Health Organization Report on Influenza Vaccine Use, http://www.who.int/influenza/vaccines/use/en/, accessed on 3 April 2023) [22–24].

Therefore, to prevent the spread and the negative consequences of seasonal influenza, the Italian Ministry of Health recommends and offers free-of-charge vaccination against these viruses to people aged 60 years and older and other risk groups (https://www.trovanorme.salute.gov.it/norme/renderNormsanPdf?anno=2022&codLeg=87997&parte=1%20&serie=null, last accessed on 3 April 2023). Despite this, the influenza vaccination program has been unsuccessful, with low rates of uptake in Italian people ≥65 years [25]. In addition, being vaccinated does not completely eliminate the risk of contracting a virus, especially by coming into close contact with other and/or unvaccinated people [26,27] such as family caregivers in the home environment. Therefore, the refusal of family caregivers to get vaccinated for seasonal influenza could have consequences, even lethal, for their relatives with dementia [28].

Given that currently no definite cure exists for dementia, family caregivers have a low expectation about the therapeutic benefits of drug treatments for dementia [29]. This aspect could also influence their general view about drugs and vaccines efficacy [28], and therefore may affect their choice to receive a seasonal influenza vaccine. Therefore, it is necessary to identify the factors that influence the family caregivers' choice to receive a seasonal influenza vaccine in order to promote more targeted vaccination campaigns.

The theory of planned behavior (TPB), proposed by Ajzen [30], represents an important theoretical model used to predict an individual's behavior in terms of intention to get vaccinated [31,32]. This theory postulates that intentions are the most proximal determinants of behavior and that three factors converge to predict intentions [33]: (i) attitudes (i.e., the psychological tendency to evaluate a particular entity as favorable or unfavorable); (ii) subjective norms (i.e., the perceived expectations of significant others and motivation to conform to these expectations in relation to a particular behavior); (iii) and perceived behavioral control (i.e., the perception of how easy or difficult it is to perform a particular

behavior) [34]. The TPB model was successfully used to predict the intention to receive a vaccine in other groups, such as the adult general population [32,35,36], older adults [37], students [38,39], pregnant women [40], and cancer patients [34]. Moreover, Ball and colleagues [41] developed a vaccine campaign based on the TPB model, namely the *Shot Talk*, which has proven useful in reducing COVID-19 vaccine hesitancy among college students.

In the context of TPB, other evidence suggests that past behaviors were also predictive of corresponding future behaviors [42–44]. More specifically, it has been shown that past history of seasonal influenza vaccine could be an additive prediction of future intention to receive it again [45–48]. Therefore, it is interesting to explore the impact of the TPB beyond past vaccination behavior.

Based on these premises, the aims of this study were to investigate the predictive role of the TPB model (i.e., attitudes towards vaccination, subjective norms, and perceived behavioral control) and past vaccination behavior on the intention to receive a seasonal influenza vaccine among family caregivers of people with dementia. We hypothesize that lower negative attitudes towards vaccines, higher subjective norms, and higher perceived behavioral control predict the intention to receive a seasonal influenza vaccine among family caregivers of people with dementia (H1). Past seasonal influenza vaccination behavior could also add to the understanding of intention beyond the TPB model (H2).

## 2. Materials and Methods

### 2.1. Participants and Procedure

Data were obtained from seventy-one family caregivers of patients with dementia. Contact information on family caregivers was obtained from the Regional Neurogenetic Centre—ASPCZ (Lamezia Terme, Catanzaro, Italy). The CRN aims to carry out research and provide assistance to people affected by neurodegenerative diseases. In its almost 30 years of activity, it has become a point of reference for patients and family caregivers, not only for the Calabrian provinces, but also for other Italian regions (see http://www.arn.it/it/crn/, accessed on 3 May 2023) [49]. To perform the present study, a cross-sectional web-based survey design was adopted to limit face-to-face contacts due to the COVID-19 pandemic, using the free software Google Forms®. The online survey was distributed between July and September of 2021. We used a forced response strategy to avoid missing data. An informational letter about the purpose of the study was mailed to all participants, along with a link to the questionnaires. Individuals were informed that participation in the study was voluntary, the survey was anonymous, and they could withdraw from the study at any time. The study was conducted in accordance with the Declaration of Helsinki and approved by the Ethical Committee of Calabria Region (Catanzaro, Italy; protocol code 52098, 16 April 2021).

### 2.2. Measures

Socio-demographics and clinical factors. Questions about sociodemographic characteristics were asked to family caregivers at the end of the survey. Specifically, participants reported the gender, age, educational level, marital status, employment status, and economic condition of the family member with dementia, as well as themselves. Family caregivers also answered questions related to the clinical conditions of their family members affected by dementia (i.e., type of diagnosis, year of diagnosis, and current disease stage).

Attitudes towards vaccination. Attitudes toward vaccination were evaluated using the Italian version of the Vaccination Attitudes Examination Scale (VAX-I scale) [50]. It consists of 12 items which can be divided into four subscales (mistrust of vaccine benefit, worries about unforeseen future effects, concerns about commercial profiteering, and preference for natural immunity), each indicated by three items. All items are presented in the form of a statement, with responses on a 6-point Likert-type scale ranging from 1 (strongly disagree) to 6 (strongly agree). Higher scores on each subscale reflect stronger anti-vaccination attitudes. In our sample, internal consistency was good; Cronbach's was: $\alpha = 0.89$ for

mistrust of vaccine benefit, $\alpha = 0.71$ for concerns about commercial profiteering, $\alpha = 0.88$ for preference for natural immunity, and $\alpha = 0.91$ for worries about unforeseen future effects.

Subjective norms. Subjective norms were measured using six items rated on a 5-point Likert-type scale, ranging from 1 (strongly disagree) to 5 (strongly agree), adapted from a previous study [34]. An example item is "People who are important to me want me to have seasonal influenza vaccine". Items were averaged to create the composite, with higher scores reflecting norms favoring vaccination. The scale showed an adequate internal consistency $\alpha = 0.70$.

Perceived behavioral control. Perceived behavioral control was measured with four items rated on a 5-point Likert-type scale ranging from 1 (strongly disagree) to 5 (strongly agree) adapted from a previous study [34]. A sample item is "I am confident that I will be able to easily get the seasonal vaccine when it becomes available". Items were averaged to create the composite, with higher scores reflecting greater confidence in one's ability to obtain the vaccine. The scale showed a good internal reliability ($\alpha = 0.82$).

Past vaccination behavior. Respondents were asked whether they had received any seasonal influenza vaccination in the past three years (yes/no).

Intention to get a seasonal vaccine. Intention to get a seasonal vaccine was measured by asking participants the following question: "I intend to have the anti-seasonal vaccine when it becomes available" (yes/no), according to a previous study [28].

### 2.3. Statistical Analysis

Data were analyzed in the Jamovi software (version 2.3.18). Descriptive statistics were used for demographic and clinical variables. To explore and identify the factors related to the intention to receive a seasonal influenza vaccine, correlations followed by a hierarchical binary logistic regression analysis were used, with odds ratios (ORs) and 95% confidence intervals (CIs). "The intention to receive a seasonal influenza vaccine" was entered as the outcome variable, and predictors were selected a priori based on their correlations with the criterion (concerns about commercial profiteering, subjective norms, perceived behavioral control and past vaccination behavior). These variables were divided into two blocks: TPB variables were entered into the first block, and the past vaccination behavior into the second block, according to the theoretical background discussed above.

### 3. Results

*Participants Characteristics*

Descriptive characteristics of the respondents and their familial affected by dementia are provided in Tables 1 and 2, respectively. The sample was composed of seventy-one family caregivers of people with dementia (70.4% female, mean age: $55.9 \pm 12.5$). Most of these were unemployed (53.5%), had a low level of school education (66.2%), and standard economic conditions (54.9%). Most patients were affected by Alzheimer's disease (67.6) followed by frontotemporal dementia (16.9%).

It is evident from Table 3 that the intention to receive the seasonal influenza vaccine among family caregivers was significantly associated with concerns about commercial profiteering, $r = -0.285$ *, $p < 0.05$—with family caregivers who were more concerned about commercial profiteering less likely to be vaccinated—subjective norms, $r = 0.323$ **, $p < 0.01$—with family caregivers with higher subjective norms more likely to be vaccinated, perceived behavioral control, $r = 0.362$ **, $p < 0.01$—with family caregivers with higher perceived behavioral control more likely to be vaccinated, and past vaccination behavior $r = 0.790$, $p < 0.001$—with family caregivers who received a seasonal influenza vaccine in the previous three years more likely to be vaccinated.

**Table 1.** Demographic characteristics of family caregivers.

| Variable | Categories | Frequency | Percentage |
|---|---|---|---|
| Age, mean (SD) | 55.9 (12.5) | - | - |
| Gender | Males | 21 | 29.6 |
| | Females | 50 | 70.4 |
| Marital status | Single | 8 | 11.3 |
| | In a relationship | 63 | 88.7 |
| Education | Low (<13 years) | 47 | 66.2 |
| | High (>13 years) | 24 | 33.8 |
| Occupation | Employed | 33 | 46.5 |
| | Unemployed | 38 | 53.5 |
| Economic conditions | Extremely problematic | 2 | 2.8 |
| | Some problems | 15 | 21.1 |
| | Standard conditions | 39 | 54.9 |
| | Medium–high | 15 | 21.1 |

**Table 2.** Demographics and clinical characteristics of patients.

| Variable | Categories | Frequency | Percentage |
|---|---|---|---|
| Age, mean (SD) | 75.5 (10.4) | - | - |
| Gender | Males | 20 | 28.2 |
| | Females | 51 | 71.8 |
| Marital Status | Single | 28 | 52.5 |
| | In a relationship | 43 | 47.8 |
| Education | Low (<13 years) | 65 | 91.5 |
| | High (>13 years) | 6 | 8.5 |
| Diagnosis | Alzheimer's Disease (AD) | 48 | 67.6 |
| | Frontotemporal Dementia (FTD) | 12 | 16.9 |
| | Dementia with Lewy bodies (DLB) | 2 | 2.8 |
| | Vascular Dementia (VD) | 6 | 8.5 |
| | Mixed (AD + VD) | 3 | 4.2 |
| Disease stage | Low grade | 9 | 12.7 |
| | Moderate | 38 | 53.5 |
| | Severe | 24 | 33.8 |

**Table 3.** Correlation matrix with all potential predictor variables.

| | | 1 | 2 | 3 | 4 | 5 | 6 | 7 |
|---|---|---|---|---|---|---|---|---|
| 1 | Intention to receive seasonal influenza vaccine | — | | | | | | |
| 2 | Mistrust of vaccine benefit | 0.227 | — | | | | 0.152 | 0.133 |
| 3 | Worries about unforeseen future effects | −0.125 | −0.043 | — | | | 0.171 | 0.071 |
| 4 | Concerns about commercial profiteering | −0.285 * | −0.299 * | 0.537 *** | — | | −0.109 | −0.062 |
| 5 | Preference for natural immunity | −0.070 | −0.275 * | 0.387 *** | 0.596 *** | — | 0.021 | 0.02 |
| 6 | Subjective norms | 0.323 ** | 0.152 | 0.171 | −0.109 | 0.021 | — | |
| 7 | Perceived behavior control | 0.362 ** | 0.133 | 0.071 | −0.062 | 0.02 | 0.189 | — |
| 8 | Past vaccination behavior | 0.373 ** | 0.013 | −0.003 | −0.065 | −0.012 | 0.031 | 0.194 |

Note: * Significant at 0.05 level. ** Significant at 0.01 level. *** Significant at 0.001. Pearson correlations were used for associations between one continuous and one dichotomous variables; phi coefficient was used for the association between two dichotomous variables.

A hierarchical binary logistic regression analysis was conducted to determine the influence of TPB variables and past vaccination behavior on the intention to receive a seasonal influenza vaccine (Table 4). Our first model, which included TPB variables (Table 4; model 1), i.e., attitudes towards vaccine (i.e., concerns about commercial profiteering), perceived behavioral control, and subjective norms, explained 51.6% of the variance in intention to receive a seasonal influenza vaccine (adjusted $R^2$ = 0.516). According to this model, concerns about commercial profiteering were negatively associated with the intention to receive seasonal influenza vaccine (OR = 0.73, 95% CI 0.55–0.96). Conversely, we have found a positive association between the intention to receive the seasonal influenza

vaccine and subjective norms (OR = 1.27, 95% CI 1.01–1.61) and perceived behavioral control (OR = 1.66, 95% CI 1.0–2.51). The second model considered in this study, which included TPB variables, as well as the past vaccination behavior (Table 4; model 2), explained 58.8% of the variance in intention to receive a seasonal influenza vaccine (adjusted $R^2$ = 0.588). Past vaccination behavior was associated with a 2.28-times greater likelihood of receiving a seasonal influenza vaccine (OR = 2.28, 95% CI 1.01–5.13). Past vaccination behavior, concerns about commercial profiteering, subjective norms, and perceived behavioral control were all significant predictors of the intention to receive a seasonal influenza vaccine.

**Table 4.** Hierarchical binary logistic regression analysis with intention to receive a seasonal influenza vaccine as outcome variable.

| Predictors | β | SE β | Odds Ratio | Z | Wald Statistic | df | *p*-Value | 95% CI Upper | Lower |
|---|---|---|---|---|---|---|---|---|---|
| Model 1 | | | | | | | | | |
| Concerns about commercial profiteering | −0.31 | 0.14 | 0.73 | −2.20 | 7.47 | 1 | 0.028 | 0.55 | 0.96 |
| Subjective norms | 0.24 | 0.11 | 1.27 | 2.08 | 5.78 | 1 | 0.038 | 1.01 | 1.61 |
| Perceived behavioral control | 0.50 | 0.21 | 1.66 | 2.38 | 10.23 | 1 | 0.017 | 1.09 | 2.51 |
| Model 2 | | | | | | | | | |
| Concerns about commercial profiteering | −0.29 | 0.14 | 0.74 | −2.03 | 6.42 | 1 | 0.042 | 0.55 | 0.99 |
| Subjective norms | 0.22 | 0.11 | 1.25 | 2.03 | 5.21 | 1 | 0.043 | 1.00 | 1.56 |
| Perceived behavioral control | 0.48 | 0.22 | 1.62 | 2.18 | 7.67 | 1 | 0.029 | 1.05 | 2.50 |
| Past seasonal influenza vaccine behavior | 0.82 | 0.41 | 2.28 | 1.99 | 4.52 | 1 | 0.047 | 1.01 | 5.13 |

Model 1: Deviance = 38.4, AIC = 46.4, BIC = 55.4, McFadden's Pseudo $R^2$ = 0.405, Cox and Snell $R^2$ = 0.308. Nagelkerke $R^2$ = 0.516, Overall model test $\chi^2$ = 26.1 (df = 3, $p < 0.001$). Model 2: Deviance = 33.8, AIC = 43.8, BIC = 55.2, McFadden's Pseudo $R^2$ = 0.475, Cox and Snell $R^2$ = 0.351. Nagelkerke $R^2$ = 0.588, Overall model test $\chi^2$ = 30.7 (df = 4, $p < 0.001$).

## 4. Discussion

To our knowledge, this study represents the first attempt in the international literature to identify factors associated with the intention to receive a seasonal influenza vaccine among family caregivers of people with dementia. Based on previous findings on other populations [34,43,45], we hypothesized that the TPB model and past history of seasonal influenza vaccines would predict the intention to receive a seasonal influenza vaccine also in our subject group. The analysis confirmed the hypothetical–theoretical model. Results of the hierarchical logistic regression showed that TPB (i.e., attitudes towards vaccination, subjective norms, and perceived behavioral control) explained 51.6% of the variance in intention to receive a seasonal influenza vaccine. Accordingly, it has been reported that the TPB model explained 51.7% [38] and 44% [35] of variance in seasonal vaccine intentions among college students and the UK adult general population, respectively.

In addition, our results demonstrated that the extension of the TBP model with past vaccination behavior increased the explained variance in the intention to receive a seasonal influenza vaccine among family caregivers of people with dementia to 58.8%. In particular, this intention was positively associated with past seasonal influenza vaccine behavior, followed by perceived behavioral control and subjective norms, and negatively associated with concerns about commercial profiteering. These findings suggested that it could be helpful to involve family caregivers who have received a seasonal influenza vaccine in previous years in vaccination campaigns, to share directly with their peers the advantages and disadvantages associated with this behavior. As mentioned above, concerns about commercial profiteering were negatively associated with the intention to receive a seasonal influenza vaccine, indicating that family caregivers who were more wary of the influence of the powerful pharmaceutical companies in the development and deployment of vaccines are less intentioned to get a seasonal influenza vaccine. These results are in line with

the study of Paul and colleagues [51], who found that this attitude was one of the most important determinants of uncertainty and unwillingness to vaccinate against COVID-19 in a large sample of UK adults.

Future vaccination campaigns might try to address the concerns about economic profits to increase the rate of seasonal vaccination in family caregivers of people with dementia. Moreover, the *Shot Talk* campaign proposed by Ball and colleagues [41] has been found to be useful in increasing perceived behavioral control and subjective norms and in improving attitudes towards vaccines among college students. Future research should evaluate the effectiveness of such a campaign to reduce the rate of vaccine hesitancy among family caregivers of people with dementia in the perspective of the TPB model.

This research has several limitations. First, it is necessary to underline that the use of a convenience sample has potentially increased the likelihood of bias. In future research, it would be more appropriate to use random sampling techniques to enhance the internal validity of the study. Second, the dimensions of this study were assessed using self-reported measures. Future research should take into consideration different methods to reduce the influence of self-report bias. Finally, a longitudinal design would be ideal to understand whether the TPB model also predicts the shift of vaccine intention into behavior in this population.

## 5. Conclusions

To decrease the negative consequences of seasonal influenza in people with dementia, it could be useful to promote seasonal influenza vaccinations also amongst their family caregivers, with whom they spend most of their time. Past vaccination behavior and the theory of planned behavior variables effectively predict influenza vaccine willingness in family caregivers of people with dementia and should be targeted in vaccination campaigns.

**Author Contributions:** Conceptualization, F.B. and P.A.; methodology, F.B. and P.A.; formal analysis, F.B. and P.A.; data collection, V.L., R.C., S.M.C., F.F. and G.P.; data curation, F.B. and P.A.; writing—original draft preparation, F.B. and P.A.; writing—review and editing, R.D.L., A.C.B. and R.M. All authors have read and agreed to the published version of the manuscript.

**Funding:** This research received no external funding.

**Institutional Review Board Statement:** The study was conducted in accordance with the Declaration of Helsinki and approved by the Ethical Committee of Calabria Region (Catanzaro, Italy; protocol code 52098, 16 April 2021).

**Informed Consent Statement:** Informed consent was obtained from all subjects involved in the study.

**Data Availability Statement:** The data presented in this study are available on request from the corresponding author.

**Conflicts of Interest:** The authors declare no conflict of interest.

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
