# Peer review of "Using the Theory of Planned Behavior and Past Behavior to Explain the Intention to Receive a Seasonal Influenza Vaccine among Family Caregivers of People with Dementia"

_2673-8937, doi:10.3390/ijtm3020017_

Round 1

Reviewer 1 Report

The reviewer suggests to eliminate the phrase “Italy showed a higher excess mortality attributable to seasonal influenza when compared to other European countries, and especially in the elderly”, because it does not refer to people with dementia but elderly population and as authors stated, dementia not only is associated to age but other causes that can be present in patients that are not considered elder (secondary dementia).

In methods section

The reviewer considers necessary to explain further about the context of the medical center in which the research was performed. Details about the influence area of operation of the clinic in terms of population attended, social security status of the population or if it has universal access. These details may aim to a deeper understanding of the research context and also its representativeness.

Additionally, the reviewer consider necessary to explain further how the levels of the hierarchical model was set because it is not clear how data was clustered to define how observations was nested and analyzed.

It is also necessary to report fitness of the models and not only the amount of variance is explained by the model.

In results section

The reviewer suggests enriching this work with a visualization of the correlation matrix. A correlogram would be the reviewer recommendation.

In another note in the results, authors explain the variance of the models using r2 estimates, however, this is correct for linear regression but not for logistic regression. McFadden psuedor estimation would be correct for logistic models.

Please avoid writing errors like the next one.

As an example “It is evident from Table 3 that the intention to receive seasonal influenza vaccine  among family caregivers was significantly associated with concerns about commercial profiteering r = -0.285*, p<0.05 - with family caregivers who were more concerns  about commercial profiteering less likely to be vaccinated.

Minor editing of english language required.

Author Response

Comment 1. The reviewer suggests to eliminate the phrase “Italy showed a higher excess mortality attributable to seasonal influenza when compared to other European countries, and especially in the elderly”, because it does not refer to people with dementia but elderly population and as authors stated, dementia not only is associated to age but other causes that can be present in patients that are not considered elder (secondary dementia).

Response 1.  Thank you to the reviewer for this suggestion. We have now delated this specification.

Comment 2. The reviewer considers necessary to explain further about the context of the medical center in which the research was performed. Details about the influence area of operation of the clinic in terms of population attended, social security status of the population or if it has universal access. These details may aim to a deeper understanding of the research context and also its representativeness.

Response 2. Thanks to the reviewer for this comment. we have now added more information about our center.

Comment 3. Additionally, the reviewer consider necessary to explain further how the levels of the hierarchical model was set because it is not clear how data was clustered to define how observations was nested and analyzed.

Response 3. Thanks to the reviewer for this comment. the levels of the hierarchical regression are defined by the theoretical framework (TPB and TPB plus past history of seasonal influenza vaccine). We have now added this specification in the statistical analysis (marked in red).

Comment 4. It is also necessary to report fitness of the models and not only the amount of variance is explained by the model.

Response 4. Thanks to the reviewer for this comment. In addition to Cox & Snell R2 and Nagelkerke R2 we have now added other model fit measures (i.e., Deviance, AIC, BIC =55.4, McFadden's Pseudo R2, Overall model test χ²) (see table 4).

Comment 5. The reviewer suggests enriching this work with a visualization of the correlation matrix. A correlogram would be the reviewer recommendation.

Response 5. Thank you to the reviewer for this comment. We have now added a complete correlation matrix (see table 3).

Comment 6. In another note in the results, authors explain the variance of the models using r2 estimates, however, this is correct for linear regression but not for logistic regression. McFadden psuedor estimation would be correct for logistic models.

Response 6. Thank you to the reviewer for this appreciated comment. Both Cox & Snell R2 and Nagelkerke R2 are widely used in logistic regression (see: Allison, 2012. Logistic regression using SAS: Theory and application. SAS institute). However, we have now added McFadden's Pseudo R2 for both model (marked in red). We have explained the variance of the models using Nagelkerke r2 accordingly to several previously publications (e.g., Askari, Marjan, Nicky Sabine Klaver, Thimon Johannes van Gestel, and Joris van de Klundert. "Intention to use medical apps among older adults in the Netherlands: cross-sectional study." Journal of medical Internet research 22, no. 9 (2020): e18080; Kapoor, K., Dwivedi, Y. K., & Williams, M. D. (2013). Role of innovation attributes in explaining the adoption intention for the interbank mobile payment service in an Indian context. In Grand Successes and Failures in IT. Public and Private Sectors: IFIP WG 8.6 International Working Conference on Transfer and Diffusion of IT, TDIT 2013, Bangalore, India, June 27-29, 2013. Proceedings (pp. 203-220). Springer Berlin Heidelberg; Constantin, C. (2015). Using the Logistic Regression model in supporting decisions of establishing marketing strategies. Bulletin of the Transilvania University of Brasov. Economic Sciences. Series V8(2), 4; Abdulqader, Q. M. (2017). Applying the binary logistic regression analysis on the medical data. Science Journal of University of Zakho5(4), 330-334, etc.).

Comment 7. Please avoid writing errors like the next one. As an example “It is evident from Table 3 that the intention to receive seasonal influenza vaccine  among family caregivers was significantly associated with concerns about commercial profiteering r = -0.285*, p<0.05 - with family caregivers who were more concerns  about commercial profiteering less likely to be vaccinated.

Response 7. Thank you to the reviewer for this comment. We have now corrected this and other grammatical errors.

Reviewer 2 Report

Dear Authors

Many thanks for the chance to read this interesting manuscript. The manuscript tests a theoretical framework (TPB) in predicting the uptake intention of a vaccination (seasonal influenza vaccination). Overall, the manuscript was methodologically adequately done. Authors had also clearly articulated theory-based hypotheses.

Abstract - The abstract would benefit from adding the most important conclusions from the study, as the implications are difficult to follow from the results.

Introduction - Considering the emphasis of the vaccine uptake messaging for the family care givers in the discussion (important as such), this should be better integrated in the introduction. The authors should consider making it clearer how the information from the tested hypotheses is linked with the field of vaccine uptake messaging. Also, please consider adding some relevant literature about vaccine uptake messaging.

Methods – it is clear that it is not possible to change e.g. questionnaire design afterwards – but please could the authors add few explanations for their choices. It was notable that for some of the questions the participants were asked a direct yes / no question. Why this was chosen instead of e.g. a Likert scale to probe the strength of intention (from definitive no to definitive yes). Implications of this should be considered in limitations. From theoretical point of view (also in TPB) it is recognized that strength of intention is one of the best predictors – but this does not mean a binary choice.

Statistical methods – please could the authors include a sentence about analyzing the demographic data. Also, it would be good to have some information about any missing data and how this was handled. Did authors consider any adjustments due to multiple testing?

Discussion – Overall, the discussion appears limited. Please could the authors avoid just repeating the results (especially in numeric format) in the discussion. The discussion would benefit from better integration of the previous literature from the topic and reflection the results in the background of the attempts to improve the vaccination uptake. Considering the very specific population included in here – how do the results advance theory-based understanding of the vaccine uptake among family care givers?

General - Some of the formal aspects of the manuscript such like language and referencing would need to be thoroughly checked before publication. In my view, the language at parts of the manuscript was difficult to follow, and would benefit from a language check.

General - Some of the formal aspects of the manuscript such like language and referencing would need to be thoroughly checked before publication. In my view, the language at parts of the manuscript was difficult to follow, and would benefit from a language check.

Author Response

Comment 1. Abstract - The abstract would benefit from adding the most important conclusions from the study, as the implications are difficult to follow from the results.

Response 1. Thank you to the reviewer for this comment. We have now added the conclusion in the abstract.

Comment 2. Introduction - Considering the emphasis of the vaccine uptake messaging for the family care givers in the discussion (important as such), this should be better integrated in the introduction. The authors should consider making it clearer how the information from the tested hypotheses is linked with the field of vaccine uptake messaging. Also, please consider adding some relevant literature about vaccine uptake messaging.

Response 2. Thank you to the reviewer for this comment. As this represents the first study on this topic in this specific population, we have no way to integrate the introduction with previous literature. However, we have now added a sentence about the field of vaccine uptake messaging.

Comment 3. Methods – it is clear that it is not possible to change e.g. questionnaire design afterwards – but please could the authors add few explanations for their choices. It was notable that for some of the questions the participants were asked a direct yes / no question. Why this was chosen instead of e.g. a Likert scale to probe the strength of intention (from definitive no to definitive yes). Implications of this should be considered in limitations. From theoretical point of view (also in TPB) it is recognized that strength of intention is one of the best predictors – but this does not mean a binary choice.

Response 3. Thanks for this comment. we could not have used binary logistic regression with one continuous variable. we have chosen to ask the question dichotomously in accordance with hundreds of other publications on vaccination intention (e.g., doi: 10.3389/fpsyg.2022.923316, doi: 10.3389/fpubh.2022.832444, https://doi.org/10.1007/s10389-021-01677-w).

Comment 4. Statistical methods – please could the authors include a sentence about analyzing the demographic data. Also, it would be good to have some information about any missing data and how this was handled. Did authors consider any adjustments due to multiple testing?

Response 4. Thanks for this comment. In each research there is a description of the reference sample. We have no missing data since the answer was mandatory for all questions (forced response strategy). We have added this information to the manuscript (marked in red).

Comment 5. Discussion – Overall, the discussion appears limited. Please could the authors avoid just repeating the results (especially in numeric format) in the discussion. The discussion would benefit from better integration of the previous literature from the topic and reflection the results in the background of the attempts to improve the vaccination uptake. Considering the very specific population included in here – how do the results advance theory-based understanding of the vaccine uptake among family care givers?

Response 5. Thank you to the reviewer for this comment. As this represents the first study on this topic in this specific population, we have no way to integrate the discussions with previous literature.

Comment 6. General - Some of the formal aspects of the manuscript such like language and referencing would need to be thoroughly checked before publication. In my view, the language at parts of the manuscript was difficult to follow, and would benefit from a language check.

Response 6. Thanks for this comment. we have now adjusted the formal aspects. For the editing of the language, we will use the service offered by MDPI if the article is accepted for publication.

Round 2

Reviewer 2 Report

Dear Authors

Many thanks for the revised manuscript. I have carefully read your revision and feedback. I do apologisise, if my previous comments haven’t been clear enough. However, I feel that the authors haven’t sufficiently engaged with the comments regarding introduction and discussion. Undoubtedly, the authors have researched an unique population. However, that doesn’t mean that literature about TPB and vaccine uptake among other populations cannot be included – there is a plenty of literature about TPB and vaccine uptake. This would help the reader to place their work among this unique population better in context of vaccine uptake research. Did the results among this population reflect similarities / differences among other population? Did the TPB predict influenza vaccine intake similarly than among other studied populations? What were the defining differences in here or did the TPB similarly predict vaccine uptake than in previous studies - even though the studied population in here was unique? What are the implications of these results in understanding TPB in connection of vaccine uptake message planning? Different messaging needs?

Regarding the statistical analyses – I didn’t quite understand the whole replay. My apologies. Thank you for adding the sentence about forced data entry methods. No mention was made about the authors considering adding a sentence about how demographic variables were analyzed. Further, authors, quite correctly, indicated that their decision of using binary logistic regression informed binary data collection for the intention variable. Please could the authors consider making it clear that binary – not multiple logistic regression was used.

Accept authors replay.

Author Response

Dear reviewer,

Thank you for your reply. We have now added in the introduction some previous studies about the use of TPB to predict the intention to receive a seasonal influenza vaccine in other populations. We felt it was inappropriate to include vaccine uptake studies as requested by you, given that our study variable is intention and not behavior (i.e., uptake).  in discussion we have now added the results of two studies which agrees with our findings. In the discussions there are already suggestions on how vaccination campaigns should be set up in this specific category of subjects based on our results and suggestions on future study perspectives (for example, the Shot Talk campaign proposed by Ball and colleagues could be useful in increasing perceived behavioral control and subjective norms and in improving attitudes towards vaccines among family caregivers of people with dementia (lines 252-273).

Regarding the statistical analysis, we have now added a sentence about the use of descriptive statistics for demographical and clinical variables. We have never mentioned multiple logistic regression in the manuscript but hierarchical logistic regression. We have now added the "binary" specification.

Round 3

Reviewer 2 Report

Dear authors

Many thanks for engaging with the comments. While not quite agreeing with your reasoning - I can - up to a point - understand it. However, in my view this leaves the work somewhat inadequately imbedded in the literature. At the moment reader is clear about the paper examining variables according to the TPB that are associated with intention to vaccinate within a specific population. However - how this relates to wider vaccination literature - e.g. improving designs of vaccine uptake programmes using TPB - is not included in the introduction - but alluded in the discussion. This could have been achieved with a sentence or two.

However, considering your feedback about the pending english language - minor corrections this time around regarding the language.

moderate editing.

Author Response

Thanks to the reviewer for this comment. As far as we know, the only vaccine campaign program based on the TPB model that has been tested is the shot talk (already mentioned in the discussions). As you requested, we have added a sentence in the introduction about this vaccination campaign. If you believe that the manuscript could be further enriched or improved, please indicate more specifically the reference literature (e.g. DOIs). While we will be using the MDPI language editing service, we also asked a native speaker researcher to point out grammar errors and he found none in the current version of the manuscript. Can you tell us exactly what they are? Thank you